# Adapting and Accepting Type 1 Diabetes: A Qualitative Exploration of the Perspectives from People with Type 1 Diabetes from 13 Countries

**DOI:** 10.3390/healthcare13121380

**Published:** 2025-06-09

**Authors:** Elsa Gaillard, David Beran

**Affiliations:** 1Institute of Global Health, Faculty of Medicine, University of Geneva, 1202 Geneva, Switzerland; 2Division of Tropical and Humanitarian Medicine, Faculty of Medicine, University of Geneva and Geneva University Hospitals, 1211 Geneva, Switzerland

**Keywords:** type 1 diabetes, adaptation, acceptance, psychosocial factors

## Abstract

**Introduction:** Divergent theories exist concerning the constructs of adaptation and acceptance for chronic conditions. Understanding these processes is essential to improving care. The aim of this study is to describe these concepts from the perspective of people living with type 1 diabetes. **Methods:** A secondary analysis was conducted on data from a qualitative study, including interviews with 101 people in 13 countries across all continents, with participants of varying ages, genders, and diabetes durations as well as participants who were parents with children with type 1 diabetes. The initial study included a topic guide with open questions and interviews were analyzed using grounded theory resulting in a pyramid of needs for type 1 diabetes. This pyramid included the concepts of adaptation and acceptance. This study explores these themes in more depth. **Results:** Adaptation and acceptance processes vary from one person to another. Adaptation includes both adjustment to daily care and a change in perspective. Acceptance is explained as a process relying on adaptation, with no defined standardized stages. Diabetes acceptance positively impacts health, daily life, and relationships. The study identifies several factors that help acceptance, such as medical supplies, a caring healthcare environment, family and peer support, parental acceptance of diabetes, and diabetes camps. **Discussion:** Some of the elements required for adaptation and acceptance can be provided directly by the healthcare system, such as medical supplies, while others outside the healthcare system still need to be considered by caregivers, such as the individual’s social environment. The psychological well-being of people with type 1 diabetes and parents should be investigated as often as possible with the provision of adaptative, integrated, and holistic care.

## 1. Introduction

Type 1 diabetes (T1D), with its onset at any age, its chronic nature, and its daily care needs, requires a variety of health system and non-health system elements for its proper management. Diabetes is managed more than 99% of the time by the individual outside the healthcare system, implying a self-management burden [1]. Psychological issues in the management of T1D play an important role in both well-resourced and low- and middle-income countries [2]. People with T1D are at a greater risk of mental health problems, such as eating disorders; depressive symptoms; and suicidal ideation, suicidal attempts, and suicide [3,4,5,6]. They also face specific mental health problems such as diabetes distress and diabetes burnout [7,8,9]. Diabetes distress is the negative psychological response to living with diabetes [10]. Depression and distress are distinct but overlapping and interrelated concepts [11]. Diabetes burnout concerns a high level of distress with significant impact on diabetes management [10]. Diabetes burnout refers to the mental and physical exhaustion of diabetes with detachment of self-management responsibilities [7,10].

Studies have reported that a lack of acceptance can have an impact on management of chronic conditions and that acceptance of T1D is linked with better management of the condition and better mental health [12,13,14,15]. Diabetes acceptance is shown to reduce diabetes distress [16]. It is essential to better understand how to help people accept diabetes in order to participate in the prevention of mental health problems.

Models of adaptation to and acceptance of chronic conditions are described in the literature. The onset of a chronic condition, such as T1D, is a biographical disruption impacting all elements of daily life [17]. People need to adapt to their new life as well as grieve their deteriorated health and create a new identity. Weinert et al. describe acceptance as the integration of the disease into the individual’s daily life [18]. Echoing this view, Babler and Strickland define accepting diabetes as integrating the disease as something normal that is part of the individual by creating routines to make diabetes part of oneself [19]. Similarly, Casier et al. make the link between adaptation and acceptance by saying that acceptance results in the recognition of the need to adapt to having a chronic condition in parallel to recognizing the various challenges posed by having such a condition [20]. From a different perspective, Stanton et al. focus only on adaptation and highlight three main elements about this process for chronic illness: chronic conditions require adjustments across a range of factors; these adjustments require time; and there is no standard pathway between individuals as they all adapt to their chronic condition differently [21]. Broadening the understanding, Oris et al. describe four illness identity dimensions related to T1D [13,22]. Engulfment is when diabetes is completely defining the identity. Rejection is when diabetes is being rejected from the identity. Acceptance is when diabetes is being accepted as part of the identity. Finally, enrichment is when diabetes is considered as an enrichment through positive changes brought on by the illness.

Bringing together the key elements of the models discussed, adaptation involves the practical and emotional adjustments individuals make to integrate diabetes into their daily lives, such as adjusting routines and learning from experience [18,19,21]. By contrast, acceptance is the process of coming to terms with the condition; acknowledging having a chronic illness and integrating it into one’s identity, without denial or letting it dominate one’s sense of self [13,18,19,20,21,22].

In a hierarchy of needs for individuals with type 1 diabetes, adaptation is identified as a prerequisite for acceptance [23]. Building off this work, this study aims to further describe the concepts of adaptation and acceptance using the same data that was collected to develop the hierarchy of needs for T1D [23]. Most existing research on adaptation to and acceptance of T1D has been conducted in high-income countries, often within single settings and guided by theoretical models. This study addresses a gap by offering a grounded, cross-cultural perspective, drawing on the experiences of both individuals with T1D and their parents across low-, middle-, and high-income countries. It both corroborates existing models and highlights factors influencing acceptance. The qualitative approach provided rich insights into lived experiences while its diverse implementation reveals shared processes and elements influencing adaptation.

## 2. Materials and Methods

Qualitative data from previous work were used to investigate descriptions of adaptation and acceptance in T1D from the perspectives of individuals [23,24]. The initial study included a topic guide with open questions and using grounded theory with the aim to generate a wide range of information on the topic of interest [25].

Theoretical sampling was used with interviews organized in 13 countries, including low-, middle-, and high-income settings. A planned target of 8 people with no limits as to their age, age at diagnosis, duration of diabetes, or sex was selected by local contacts in each country. The number of people and countries involved in the study was based on opportunistic sampling. The number of participants differed across countries due to logistical constraints and recruitment opportunities.

The interviews were conducted in the following countries (with number of interviewees): Argentina (10), Indonesia (3), Kyrgyzstan (8), Mozambique (8), Nicaragua (8), South Africa (9), Singapore (7), Switzerland (8), Thailand (8), Tanzania (8), United Kingdom (8), USA (8), and Vietnam (8), with a total of 101 interviews.

In some settings, children and adolescents were selected. These children and adolescents were interviewed with their parents present and with authorization, in a joint interview with parents and children, or with just parents. All participants were volunteers, and informed consent was obtained from all the participants. Translators were used in Indonesia, Kyrgyzstan, Thailand, and Vietnam; in all other countries, DB carried out the interviews in English, French, Portuguese, and Spanish. On average, interviews lasted 52 min over the course of one interaction. All interviews were transcribed verbatim and analyzed using NVivo software (NVivo 7; QSR International, Melbourne, VIC, Australia).

The 101 interviews included 56 females and 45 males, with an age range of 1.2 years to 61.0 years (median 22.0 years). No other socio-demographic characteristics or biological measures, e.g., HbA1c, were taken.

The overall result of this work was the hierarchy of needs for type 1 diabetes [23], with the following themes identified from the base to the top of the pyramid: ‘healthcare workers’; ‘information and education’; ‘insulin’, ‘delivery of insulin’, and ‘control’, e.g., blood or urine glucose; ‘policies’ and ‘organization of health system’; ‘community support’; ‘family support’; ‘peers’; ‘experience’ and ‘personality’; ‘adapting’; ‘being open’; ‘acceptance’; and ‘diabetes as something positive’.

Individuals were identified with a code, including their country (AR, CH, IN, KG, MZ, NIC, SA, SIN, TH, TZ, UK, USA, and VT), sex (F, female; M, male), and age; for example, NICF23 is female, age 23 years from Nicaragua.

From the original transcripts, EG extracted all mention of the terms “Adapting” and “Acceptance” by the interviewee and analyzed these in more depth using content analysis. Coding of “Adapting” was performed when the individual described how they adapted their diabetes management to a given situation and/or over time. Transcripts were coded with the term “Acceptance” when the interviewee mentioned the word “acceptance” or described factors related to or contributing to acceptance.

## 3. Results

Based on the qualitative interviews conducted to develop the hierarchy of needs for T1D, this section presents key themes that emerged using grounded theory from participants’ accounts of adapting to and accepting life with T1D to offer insight into the psychological and practical challenges of T1D.

Seventy-four (73.3%) of the one hundred one interviewees mentioned the issue of adaptation or acceptance during their interview. Of the 27 who did not mention these two elements, 22 (81.5%) came from low-income countries, and 17 (63.0%) were from interviewees under the age of 18. Adapting was mentioned by 74% of interviewees. Acceptance was described by 38%. Adaptation was coded 367 times, and acceptance was coded 144 times. The results are presented in two parts. The first explores how participants adapted to the daily realities of living with T1D. The second examines the process of acceptance and the various factors that shape and influence it.

### 3.1. Adapting to Type 1 Diabetes

Concerning adaptation to diabetes, MZM42 expressed how he “had to adapt to this reality”, with USAM25 adding that “it is a whole life change” or, as put by SAF52, “a new way of life”. CHF8’s parents stated that “we adapt, we adapt all the time”. These quotes both highlight the initial adaptation to the diagnosis of T1D as well as the ongoing aspect of this process. The areas where adapting was mentioned relate to the following:Insulin use;Varying dosage;Different types;Insulin delivery using a pump;Blood glucose meters (BGMs);Eating;Activities;The use of guessing and experience;Developing a routine.

#### 3.1.1. Using Insulin

In looking more closely at the use of insulin, some interviewees described a situation of changing insulin doses based on a scale linked with readings on a BGM. Others were self-adjusting their insulin, as described by ARM41, depending on “how my day goes based on what my blood glucose was and what I eat. I calculate what my blood glucose is at”. UKF28 said that this way of doing things was “hugely beneficial, much more flexible and it fitted in with my life”. These changes in the insulin dose were based not only on the information provided by their BGM but also, as stated by ARF29 on “what I know about myself”. This situation is contrasted to interviewees in Vietnam, Tanzania, and Mozambique, where people with T1D were not able to change the insulin dose. As reported by VTF11’s father, “it is not permissible to change the dosage. If anything is beyond the range or abnormal, we will report this to the doctor and ask for permission”.

Some interviewees described newer insulin formulations as having a positive impact on treatment, especially increasing their flexibility in their daily lives (ARF26, ARF29, ARF39, ARM27, and CHF14). For CHF24, these new insulins were a “complete liberation”, as she could now go out with her friends and not worry about insulin injections. UKM54 describes the following: “You always needed to take it 30 min before you eat. […] Whatever happened, it was hard to plan when to do it if I was going out. Whereas now I can just do it and I don’t have to constantly think about it”. Some participants also mentioned how the use of an insulin pump improved flexibility, led to better management, or just made things easier.

#### 3.1.2. The Role of Blood Glucose Monitoring

An essential tool to help people adapt is a BGM. UKM54, who has had diabetes for 50 years, described how “having the better equipment has made it [adapting to diabetes] easier”. Using a BGM as a tool for “adapting” also occurs for specific situations, such as when they do not feel well (ARM27, SAF52, SAM28, SINF6, UKM22, UKM54, and USAF46), when they travel (ARM27), when they change their meals (ARM41), and when they have special events, for example, for ARM18, USAF46, and USAM25, when they are performing intense physical activity.

#### 3.1.3. Adapting Using Guessing and Experience

In addition to the tangible elements of insulin and BGMs, two words frequently emerged in the interviews when looking at adapting: “guess” and “experience”. The terms “guess”, “guessing”, or “guesswork” were used when describing adapting insulin doses to meals. For example, UKF34 presented this as “if I had a desert with dinner, I would do an extra shot, but it would very much be guesswork”. CHF24, SINF3, UKF28, UKF24, and USAM25 also described this process of adapting as guesswork.

Regarding “experience”, THM27 stated the following: “The dosage [of insulin] will be adapted to my eating and activity and it is also based on my experience”. ARM27, ARM41, MZM42, THM39, UKM22, and UKM54 all used the term experience when relating to adjusting their management of diabetes.

#### 3.1.4. Developing a Routine

All the factors described above helped people adapt to their diabetes by developing a “routine” or “rules”, “planning ahead”, “doing things differently”, and “preparing”. (ARM41, CHM34, SINF3, UKM22, and USAM25). Concerning management on a day-to-day basis, the focus is mainly put on how insulin and measuring blood glucose fit into the day and around mealtime. ARF39, INM13’s mother, UKF35, and UKM22 described a certain routine and mentioned how some key events of daily life, e.g., waking up, eating breakfast, or going to school, lead to some aspect of managing diabetes, such as testing blood glucose or taking insulin.

Making diabetes a “habit” (USAF22) and having a “structure” (UKF35) to help manage this condition are necessary, as diabetes is “a pretty significant part of a day” (USAM25). ARF39, who has had diabetes for 31 years and only recently received an insulin pump, told us how “I still haven’t got used to not giving myself my insulin injection as I feel that something is missing”.

#### 3.1.5. Adapting and Meals

One of the main topics where adapting was mentioned was food. Some completely changed their children’s diets, for example, the parents of INM13, SINM13, VTF8, VTM2, and VTM11. Other parents were “quite strict with mealtimes and quantities” and say “at home we have a scale and we weigh each time she eats.” (CHF18).

SINM15, UKF35, UKF28, USAF22, and USAM25 all mentioned carbohydrate counting which had helped them adapt. ARF26 told of how her doctor explained “you can eat sugar you just need to adjust things. She taught me how to eat.” In the past, this was even more difficult, as related by UKM54: “My mom was given scales because you had to weigh everything then because of the diet and everything else”. People who have had diabetes for a long time (ARF39, SAF29, and USAF60) related that there were nowadays a variety of sugar free or “diabetes” foods available.

This adjusting of diabetes management to food intake needed to be learned. As described by the interviewees, healthcare workers and peers played a key role in this. Learning from others was described by SINF15: “I met other kids and I was one of the youngest. And the other kids they told me it’s OK to eat anything, but it has to be in moderation. It is OK to eat everything even chocolates and you don’t need to worry too much”.

#### 3.1.6. Adapting and Activities

Parents with children related how they needed to change the management of their child’s diabetes to birthday parties, sleepovers, and just going out (CHF8 and CHF18). Travelling, sport or special activities also mean additional planning with regards to insulin, food, and other aspects (CHM34, SINM5, USAM42, USAM55, ARM41, and SAM20).

Work is also a challenge in adapting to diabetes. ARF39 described how this had been a challenge in the past, but as she now was her own boss, she could adjust her schedule to her diabetes. CHM61 also faced this challenge due to the type of job he had: “I did not have an office job and was expending a lot of energy. Because I had this schedule where I worked 24-h days your system is perturbed”. This led him to change jobs.

Interviews show that individuals with T1D adapt continuously after diagnosis across multiple aspects of daily life, including insulin use, BCM, food, routines, and activities, often relying on personal experience and guesswork.

### 3.2. Acceptance of Type 1 Diabetes

#### 3.2.1. Non-Acceptance of Diabetes

Not accepting diabetes was expressed by a few people, all relating it to negative outcomes, diabetes distress, and diabetes burnout symptoms. SINF15 described how being “in denial” lead her to not manage her diabetes optimally. USAF22 adds that not feeling unwell was an obstacle to the treatment. “I think sometimes I push it to the back of my head because I feel OK, I feel fine, and it won’t be until the next day that it hits me and I’m out of breath […] That is when I tell myself I need to take my insulin”. INF16, stopped going for a medical consultation for 3 years because she felt there was so much to do for diabetes and that she felt hopeless. NICF10’s mother said that her daughter did not want to take responsibility for diabetes. INF26 expressed being “tired of having diabetes”, and USAF22 highlights a “lack of motivation” in diabetes care and “giving up [and] not caring about it”. Unlike for INF26, USAF22’s lack of acceptance was due to day-to-day tasks, such as “getting my daughter ready for school, or putting her in the shower, or packing the lunches, there is always something before it”. Not accepting diabetes led some of the people interviewed to miss out on certain things, such as school (VTF14).

#### 3.2.2. Acceptance Process and Adaptation

Acceptance was described as a long stepwise process by some interviewees (ARM19, CHF18, SAF52, and THF27). SINF18 expressed how by “growing up”, she accepted diabetes more and more. Acceptance is a dynamic process, as described by ARF24, as despite having had diabetes for 12 years, she said that “some days I ask why and other days I am fine”. Others described acceptance as a transition that took place the day they realized they had no other choice. Accepting diabetes sometimes came from a negative experience, such as realizing that if they did not “accept” diabetes and do something about their illness, it could lead them to the hospital or even to death (ARM19, THM27 and UKF57).

This process of adapting and normalizing life with diabetes enabled individuals to accept their diagnosis, as the practical aspects of life with T1D become more manageable over time. Indeed, realizing that diabetes was “not a big obstacle” and that the only thing was “not to eat sugar and look after yourself” was also part of the process of “accepting diabetes” (ARM19; ARM27, CHF24 and UKF57). ARM27 stated that “Diabetes is an important illness. If you do what you need to do, if you take care of yourself, you can live a normal life”. NICF23 and NICF30 said that they both had normal lives with a husband and child, but the only thing they needed to do differently was inject insulin. ARF29, CHF24, and INM13 highlighted that they felt they had a normal life and were normal people that just needed to do some extra things. Injections and other aspects of care had become routine. UKF57 also stated that improved knowledge on how to manage diabetes meant that it “became a smaller problem for my life and it became more acceptable”. THF27 described that the transition from not accepting to accepting happened when she understood that “the diseases cannot be cured, and I need to depend on insulin and medicines for life”. She challenges her own point of view, saying “some people would not use the term accepting it but getting used to it”. Acceptance of diabetes as a part of the identity was highlighted by CHF8’s parents with the anecdote of their daughter wearing an insulin pump and that “it has been years that she has been wearing it, so it is like part of her in the end. If she draws herself, she draws herself with the pump. It is something that is an integral part of her”. The three types of acceptance processes described in the interviews are summarized in Table 1.

#### 3.2.3. Enrichment

Some people explain that they could turn diabetes into having a positive impact on their lives. SINF15 says how now that she “learnt how to accept it more, it is turning from being a curse to a gift”. Positive aspects from having diabetes are linked to discipline for diabetes and life in general by CHF24 and UKF23. USAM22 adds that he is “conditioned to a much harder environment than most people” so that he is “able to handle more things. Some people express that they wanted to help other people with diabetes or “give back”, coming from different reasons from one person to another. USAM55 describes diabetes “as a blessing at that level and it has shaped my life and I hope I can make an impact”. The motivation to do so mostly came from their own experience (CHF18, SINF6, SINM5, USAF46, USAM22, and USAM25); having the time, ability, or opportunity to do this (CHM61, USAM25); and other people’s challenges (ARM19, CHF18, and SINF6’s and SINM5’s mothers). This motivation resulted in being involved in creating diabetes associations or support groups (mothers in Kyrgyzstan and THF32, USAM22), fundraising (USAM22 and USAM42), volunteering (USAM22), deciding to work in the area of diabetes because they had diabetes (USAF46, USAM47), and being counselors providing information and support to people with diabetes (ARF39, NICF15’s mother, NICF23, SAF52, SINM15, THM27, and USAF46). The positive aspects brought by diabetes are listed in Table 2.

### 3.3. Factors Influencing Acceptance

#### 3.3.1. Healthcare Workers

Interviewees described multiple factors influencing acceptance. The importance of the information provided by healthcare professionals was emphasized, particularly the messages conveyed at the time of diagnosis. USAM42 stated that at “9 years old is kind of a young age to have to wrap your head around something that is going to impact you for the rest of your life. […]. You know you are just going to have to get on with it”. People expressed the role played by healthcare workers in the acceptance process thanks to the information they provide. USAM47 mentioned that parents “need to know that their children will be OK”, and that healthcare workers have this role to play. As said by USAF60, “One of the most helpful things is that if healthcare workers are honest with people telling them that diabetes can be a very hard disease and it is a life changing disease, but if you follow what you are supposed to do [it can be managed] … it will help someone gain acceptance about having what they have”.

#### 3.3.2. Diabetes Camps

Diabetes camps were also described as a contributing factor, as they provide information and an example that people can live a normal life even though they suffer from diabetes (NICF24, INM13’s mother, CHM61’s parents, THM15’s parents, and CHF18). Camps allowed people with diabetes and parents of children with diabetes to meet and share their experiences. ARF46 described how at a diabetes camp “we were 40 children all with T1D of different ages and the basic thing they taught us was that we were normal children”. CHF18 also stated that diabetes camps helped her in the process of accepting her diabetes and that she learned from other people who had the same condition.

#### 3.3.3. Family and Social Support

Family support was identified by 13 interviewees as a key element in facilitating acceptance. (ARF24, ARF29, ARF46, ARM19, NICF23, NICF24, THF27, THM15, THM27, TZF13, UKF57, USAF60, and USAM47).

NICF23, THM15’s parents, and USAF60 stated that parents have to accept diabetes in order for children to accept it. For example, ARF24 described how it was hard to “see my mother cry because she had to inject me”. THM15’s parents described how the father accepted it right away and how with the passing of time, the mother learned to accept it and that this process was necessary for her son. NICF23 and THF27 described how the family has to help the child understand that in order to be well, they need to hurt themselves everyday with injections and finger pricks.

ARF29, ARF46, and CHF24 mentioned the role friends played in supporting them and therefore helping them accept diabetes.

#### 3.3.4. Personal Factors

Besides the support of others, ARF24 illustrated the individual’s own role, using the example that when she left home for university, she was on her own, and diabetes became her responsibility. Interviewees described how having a “strong character” (CHF24) or a “strong self-esteem” (NICF23) can play a role in accepting diabetes.

This feature of personality and not letting diabetes get you down was explained by ARF46 in the following way: “It is a chronic condition that has no cure and this is the bad news. This disease is going to die with you. The good news is you can have a normal life like a person without diabetes. You need to educate yourself, get information and not feel sick. Like with any other disease if you feel sick you have already lost part of the battle”.

Some interviewees mentioned religion and the fact that diabetes was determined by God and therefore was their destiny (ARM41, TZF13, and VTM11). Others found acceptance and hope in prayer and in their religious beliefs (NICF11, NICF30, TZF13, and USAM47). The factors helping acceptance described by the participants are listed in Table 3.

## 4. Discussion

This study shows a wide range of experiences with diabetes. It is not possible to summarize diabetes in terms of a single way of coping and accepting it. Each individual experiences a unique adaptation process that depends on multiple elements described in the interviews. Adaptation includes both adjustment to the daily use of insulin and other aspects of diabetes treatment and also the change in perspective of the person now suffering from a chronic condition. Adaptation takes time and experience. The interviews underline that the acceptance process is intimately linked to a process of adaptation, with no defined standardized stages, requiring a constant fight with round-the-clock care. Some participants described acceptance as a stepwise process. Others said they were able to accept their illness after realizing that they can have a normal life with diabetes. Others describe a turning point when they realized they had no choice. Some participants do not speak of accepting the illness but describe their experience as an adaptation. Both adaptation and acceptance have an impact on the individual’s health as well as on their daily life, like schooling, work, or interactions with peers. A lack of acceptance is linked to symptoms of diabetes distress and diabetes burnout. These interviews show that it is possible for diabetes to bring something positive, either in one’s own life, such as being more attentive to one’s health, or in the lives of others, such as the possibility of helping others with diabetes.

Acceptance was described as a process similar to those presented in other publications, corroborating theories that provide complementary perspectives. According to Stanton et al., adjustment to chronic illness requires adjustments to a range of factors, which takes time, and there is no standard between individuals, as they all adjust differently to their chronic illness [21]. The diversity of our findings ground this theory in lived experiences, supporting the idea that chronic illness adjustment is uniquely shaped by personal circumstances and requires time for experience.

Casier et al. state that acceptance results from the recognition of the need to adapt to a chronic illness along with the recognition of the different challenges posed by such a condition [20]. They found that higher levels of daily acceptance were associated with better mood and emotional well-being, supporting the idea that acceptance promotes psychological resilience in the face of chronic illness. Our findings resonate with this, as participants who described a sense of acceptance often linked it to improved emotional regulation. Babler and Strickland define accepting diabetes as integrating the disease as something normal that is part of the individual [19], which this study corroborates. However, they outline sequential steps describing a uniform progression, whereas this study presents a dynamic, individualized process with no fixed steps. Acceptance is portrayed in the present interviews as a dynamic, lifelong process that does not reach a fixed point, similar to the Oris et al. illness identity model [13,22]. The four illness identity dimensions were reflected in the interviews carried out for this study. Engulfment and rejection were described by individuals struggling to accept their diabetes, while acceptance and enrichment were expressed by those who had integrated the illness into their identity. Rassart et al. demonstrated that engulfment and rejection are associated with poorer outcomes, such as lower treatment adherence and increased diabetes distress, while acceptance and enrichment are linked to better well-being and self-management [13], patterns that appeared in the present interviews.

Among the factors enabling acceptance of T1D, participants describe various factors that can be directly addressed by the healthcare system, such as the supportive environment provided by healthcare professionals for people with T1D and their families, and tangible elements, such as insulin and BGMs. Factors external to the healthcare system must still be considered. Patients’ associations, such as diabetes camps, enabling peer exchanges helped both parents and children. Family and peer support also play an important role. Many parents described the need to adapt for their child to lead a “normal” life. Personal factors were mentioned.

While the study design does not support statistically generalizable conclusions, it allows for the identification of descriptive patterns across diverse contexts. An interesting finding is that 22 out of 27 of the participants who did not mention adaptation- or acceptance-related issues were from low-income countries. The management of T1D varies considerably across healthcare systems. When individuals in high-income countries are empowered to self-adjust insulin doses, with access to tools such as continuous glucose monitors and insulin pumps, low- and middle-income countries often rely on fixed insulin regimens managed under strict medical supervision, largely due to limited diabetes education, health infrastructures, and access to diagnostic tools [26,27,28]. Moreover, people with T1D in low-income countries have a shorter life expectancy due to limited access to diagnosis, insulin, and quality healthcare [29,30]. Further exacerbating disparities, access to mental healthcare differs greatly between high-income countries and low and middle-income countries, where mental health systems are often under-resourced or absent because of a shortage of trained professionals and insufficient funding [31]. The fact that most of the participants not mentioning adaptation and acceptance are from low-income countries can be explained by the lack of essential aspects of care that obstruct personal development [23].

Mental health conditions show highly variable prevalence rates across the world, largely influenced by cultural values [32,33]. Studies indicate that these values shape not only the causes (etiology) of mental health conditions but also their expression (phenomenology) [33]. While mental health stigma is a universal phenomenon, it manifests differently depending on cultural contexts [34]. It can be particularly harmful in societies where deviation from social norms leads to increased rejection [34,35]. As a result, psychological conditions may not be recognized through a Western medical lens nor openly expressed by those affected due to potential social consequences [33,34]. In the field of diabetes care, stigma and cultural beliefs surrounding treatment can also hinder therapeutic adherence [36]. This study highlights similarities across cultures but does not address specific intercultural differences, which are necessary to take into account to improve quality of care [37].

### 4.1. Limits of the Study

Given the qualitative nature of this study, issues of researcher bias; bias in the selection of topic; bias in sampling; bias in the collection, analysis, and presentation of data; and contextual bias due to the different cultures between the researcher and individual were all factors that impacted data collection and the initial analysis [38]. As this study included multiple languages, the concepts of adaptation and acceptance might be understood and translated differently. That said, the use of translators in four of the thirteen countries with experience in health and qualitative projects mitigated this, and in the other nine, the lead researcher was able to speak the local language. Although this study investigates elements closely related to psychology, neither researcher is a trained psychologist nor were the data collected for this purpose. Given that a convenience sample was used, the representativeness of respondents is a limitation.

These interviews were collected in 2013. Technological advances that have since benefited high-income countries and some middle-income countries are not represented in this study, such as continuous glucose monitoring or closed-loop pumps. However, these technological advancements have not changed diabetes management in many low- and middle-income settings, where insulin and related supplies are lacking and people with diabetes face weak health systems [27,28]. Recommendations for psychological and psychiatric screening and management and recent research on the psychological aspect of T1D are also not represented. However, many countries are unable to implement adequate mental health care due to weak health systems and the stigma of mental illness preventing people from seeking help, regardless of the country they live in [39,40,41]. Moreover, as this is a secondary analysis of qualitative data, the original aim of the main study was not necessarily to explore these concepts.

### 4.2. Contribution to Clinical Practice and Organization of Care

Many qualitative studies have looked at different elements of the challenges of managing T1D, but few have looked at the issues of adaptation and acceptance. These studies have been performed only in an individual country [42,43,44]. To the authors’ knowledge, this study is the first to look across different contexts as well as focus on describing adaptation and acceptance from the perspective of individuals living with diabetes. These interviews indicate that psychosocial factors directly influence diabetes management in both low- and high-income countries. This study shows that the acceptance of diabetes has a significant positive impact on disease management. It also highlights the factors that need to be considered by healthcare providers in the follow-up. This study emphasizes the need for adaptive and integrated care as well as a holistic approach for people with T1D and their families. This need is consistent across different countries and healthcare systems. This study is in line with the ISPAD guidelines that recommend psychological screening of people with diabetes and psychosocial and behavioral interventions as well as other studies showing that it is necessary to take into account how people feel about their condition rather than focusing on biological assessments [10,45,46]. There is the need to assess the acceptance of diabetes and detect symptoms of diabetes distress as often as possible. Furthermore, we recommend exploring the factors highlighted in these interviews when people struggle to accept their condition.

### 4.3. Further Direction of Research

This study contributes to the research on the psychosocial aspects associated with T1D and provides valuable insights into people’s needs regarding holistic care. Despite the growing interest in this topic, further research is needed on diabetes-specific mental health issues, such as diabetes distress and diabetes burnout [7,9]. The study should be repeated to see if technological advancements and psychological care recommendations influence the experiences of participants benefiting from these improvements. This study was not designed to compare levels of illness acceptance across healthcare systems and cultural contexts. The aim is to identify cross-context characteristics rather than analyze inter-context differences. Studies specifically designed for cross-context comparison would provide valuable insights into the socio-economic and cultural determinants of diabetes acceptance. Further research is needed using psychometric scales, such as the Diabetes Acceptance Scale (DAS), to evaluate the quality of interventions and screening in these areas [47]. Concerning psychosocial interventions, several models are currently implemented and evaluated: family-based interventions, psychosocial interventions during consultations, peer groups, individual therapy, and internet-based and digital interventions [10]. Further research is needed to tailor psychotherapies for T1D, such as the Acceptance and Commitment Therapy model [48,49]. Further studies should also be conducted on parents because of their impact on the adaptation and acceptance process as well as the mental health issues they present in relation to their child’s diabetes [50,51,52,53].

## 5. Conclusions

Both adaptation and acceptance have an impact on the individual’s health as well as on their daily life, like schooling, work, or interactions with peers. Several elements influence the acceptance of T1D, related to the healthcare system and beyond. More consideration should be given to diabetes acceptance and its influencing factors, including parents’ acceptance process during diabetes care and while conducting research into T1D diabetes. Adaptative and integrated care with a holistic approach, including a consideration of psychosocial aspects, is needed to ensure optimal management and care for people with T1D.

## Figures and Tables

**Table 1 healthcare-13-01380-t001:** The three paths to acceptance described in the interviews.

Acceptance Process	Participants
Stepwise process	ARF24, ARM19, CHF18, THF27
Transition after adaptation, when they realized they could have a “normal” life	ARM19, THF27, THM15’s parents, THM27, UKF57
Transition when they realized they had no choice	ARF29, ARF46, ARM19, ARM27, CHF24, INM13, INM30, NICF23, NICF30, THF27, THM27, UKF57

**Table 2 healthcare-13-01380-t002:** Positive aspects brought by diabetes described in the interviews.

Positive Aspects	Participants
Analytical	USAM22
More controlled and disciplined	UKF23 and USAM22
Feel more mature	CHF24
More careful with your health and diet	CHM34, SINF3, UKF23, UKF28, UKF35 and USAM42
More observant, sensitive, and respectful	CHM61
Able to handle more than other people	USAM22
Influence in career choices	UKF28, UKF34, UKF35 and USAM55
Able to help other people with diabetes	ARF39, ARM19, CHF18, CHM34, CHM61, NIC15’s mother, NICF23, SAF52, SINF6 and SINM5’s mothers, THM27, USAF46, USAM25, USAM42, USAM47, USAM55, mothers in Kyrgyzstan

**Table 3 healthcare-13-01380-t003:** Factors helping acceptance described in the interviews.

Factors Helping Acceptance	Participants
Family support	ARF24, ARF29, ARF46, ARM19, NICF23, NICF24, THF27, THM15, THM27, TZF13, UKF57, USAF60, USAM47
Healthcare workers	ARF46, CHF24, CHM61, NICF23, THM15, USAF60, USAM47
Psychologist	NICF10
Diabetes camp	ARF46, CHF18, CHM61, INM13’s mother, NICF23, NICF24, THM15’s parents
Information	ARF29, CHF24, CHM61’s parents, INM13’s mother, NICF24, THM15’s parents, USAF60
Seeing worse situations	ARF24, NICM15, TZF13’s mother
Friends	ARF29, ARF46, CHF24
Individual role	CHF24, NICF23, SINM13’s mother, THM27
Insulin analogues	CHF24
Religion	ARM41, TZF13, VTM11

## Data Availability

Data available on request.

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
