# Peer review of "Adapting and Accepting Type 1 Diabetes: A Qualitative Exploration of the Perspectives from People with Type 1 Diabetes from 13 Countries"

_healthcare, 2025, doi:10.3390/healthcare13121380_

Round 1

Reviewer 1 Report

Comments and Suggestions for Authors

Overall, this is a valuable study that could contribute significantly to improving patient care for those living with type 1 diabetes. 

The abstract should contain more valuable and specific information. Explain the method better in the abstract.

Introduction:

The introduction provides a comprehensive overview of the psychological challenges associated with Type 1 Diabetes (T1D) and sets the stage for the study. However, the structure could be improved by more clearly linking the different concepts (diabetes distress, burnout, and acceptance) and explaining their relevance to the current study. A clearer flow from the discussion of psychological issues to the study’s aim would help in guiding the reader through the background. By doing so, the introduction will provide a more seamless lead into the study’s aims and significance.

Material and methods: 

It is important to ensure that the selected participants reflect the diversity within each country. This could be a limitation, as the selection was made through local contacts, who may have introduced bias into the selection process if they did not have clear guidelines to ensure a balanced representation of age, sex, and disease duration. It would be helpful to clarify whether any effort was made to select a diverse sample within each country: variation in age, sex, disease duration, etc. Sample size: The total of 101 participants is adequate for a qualitative study, but more detail should be provided on why 8 participants were chosen per country. In some areas, the number of participants is relatively low (3 participants in Indonesia, why?), which could influence the representativeness of opinions from certain regions. The methodology could benefit from further justification on the selection of this number of participants per country. Explain better and justify how grounded theory will be carried out. The methodology appears well-designed for a qualitative study and appropriately addresses the complexity of adaptation and acceptance experiences of type 1 diabetes in an international context. However, greater clarity on sample selection, cultural differences, and data analysis could help strengthen the methodology and ensure the reliability and validity of the results.

Results: 

Although various relevant topics are introduced, it would be helpful to begin the results with a brief overview summarizing the context of data collection and the objectives of the analysis. This would better prepare the reader for the diversity of topics covered in the following sections. Including an introductory sentence that clarifies what is intended to be achieved with the results and how they are analyzed (e.g., according to the grounded theory model) would be beneficial.  

While the document presents the results in detail, there should be more reflection on how these results contribute to the construction of grounded theory. That is, how categories emerge, the relationships between them, and how these results respond to the initial research. It is very important and relevant to mention all the positive effects that diabetes has brought to them. The results present a rich diversity of experiences that could be better organized and contextualized to improve understanding and cohesion between the different sections.

Discussion:

The discussion refers to publications exploring the topic of acceptance and adaptation to diabetes. However, it would be useful to delve deeper into how the results obtained in this study either support or challenge previous theories. For example, it is mentioned that acceptance is a dynamic and continuous process, similar to the illness identity model by Oris, it would be interesting to see how this perspective aligns with the study’s findings and how clearer connections can be made between the results and the theories proposed by other authors. While the discussion includes some limitations, such as cultural bias and the lack of representation of certain recent technologies, you could emphasize other possible limitations of the study. The discussion could more clearly highlight the strengths of the study, such as its qualitative approach, the multidimensional context of the interviews across different countries and cultures, and its contribution to the understanding of acceptance and adaptation in various socioeconomic settings.

The bibliography is correct and up to date.

It is a well-presented and well-conducted piece of work.

Reviewer 2 Report

Comments and Suggestions for Authors

The introduction outlines the importance of adaptation and acceptance, it could better emphasize the specific gap this study aims to fill. How does it expand on previous work, and what unique contribution does it make? The terms “adaptation” and “acceptance” are discussed extensively, but the distinction between them could be made clearer. Consider summarizing key differences in a more structured manner. Some sections (e.g., the discussion of illness identity dimensions) feel fragmented. A more seamless transition between different psychological models would improve readability.

The study uses theoretical and opportunistic sampling. It would be helpful to justify how this approach ensures representation and minimizes bias. Lack of Demographic and Clinical Data: While the study reports participant age and gender, other socio-demographic and clinical characteristics (e.g., HbA1c levels, socioeconomic status) are missing. Providing such details would improve the ability to contextualize findings.

Since translators were used in some countries, it would be useful to discuss potential limitations related to language differences and how these were addressed to ensure data reliability. The methods mention NVivo 7, an outdated version of the software. If a more recent version was used, it should be updated accordingly.

The results section provides a thorough exploration of adaptation and acceptance, it would benefit from clearer subheadings and transitions between key themes. Consider explicitly linking adaptation and acceptance with supporting subthemes. The percentage of interviewees discussing adaptation and acceptance is valuable, but ensure consistency in presenting numbers. The phrase “74% of interviewees 367 times” is unclear. Does this mean adaptation was mentioned 367 times in total? Clarify how these numbers were calculated. The differences in adaptation strategies across countries (e.g., self-adjusting insulin vs. strict medical oversight) could be further explored. Including a brief comparative discussion on systemic healthcare differences would strengthen the analysis. Direct quotes add valuable insight but should be more concisely integrated to maintain readability. Consider summarizing some viewpoints rather than including multiple similar quotes. The transition between adaptation strategies and acceptance is somewhat abrupt. A brief concluding sentence for adaptation before introducing acceptance would improve flow. Ensure that key terms like “adaptation” and “acceptance” are used consistently and avoid redundancy in describing processes.

The discussion effectively highlights the individualized nature of adaptation and acceptance in diabetes management. However, the section could benefit from a clearer integration of theoretical models on chronic illness adaptation to strengthen the argument. Stanton et al. and Casier et al. are referenced, a more explicit comparison of findings with existing frameworks would enhance the discussion. The role of socioeconomic disparities in adaptation and acceptance is briefly touched upon but warrants further elaboration, particularly in the context of access to psychological care and diabetes-related technologies. Addressing these aspects would improve the study’s depth and applicability across different healthcare settings.

Comments on the Quality of English Language

The English could be improved to more clearly express the research.

Round 2

Reviewer 2 Report

Comments and Suggestions for Authors

Dear Authors, 

I am pleased with the corrections and revisions made by the authors in response to the feedback provided. The authors have effectively addressed the concerns raised, demonstrating a clear understanding of the issues and a commitment to improving the manuscript's quality.

Thank you.